# Development and Optimization of Black Rice (*Oryza sativa* L.) Sourdough Fermented by *Levilactobacillus brevis* LUC 247 for Physicochemical Characteristics and Antioxidant Capacity

**DOI:** 10.3390/foods12071389

**Published:** 2023-03-24

**Authors:** Syue-Fong Lai, Yi-Wen Chen, Shin-Mei Lee, Hsin-Yu Huang, Yu-Hsin Huang, Ying-Chen Lu, Chih-Wei Chen

**Affiliations:** 1Department of Food Science, National Chiayi University, Chiayi 60004, Taiwan; 2Department of Food Science, National Quemoy University, Kinmen 892009, Taiwan; 3Department of Long-Term Care and Management, Wu Feng University, Minxiong 621, Taiwan; 4Bachelor Degree Program in Food Safety, Hygiene and Laboratory Science, Chang Jung Christian University, Tainan City 711, Taiwan

**Keywords:** black rice, sourdough, *Levilactobacillus brevis*, anthocyanin, antioxidant

## Abstract

This study used *Levilactobacillus brevis* LUC 247 to ferment black rice sourdough, made into Type III black rice sourdough powder to produce black rice sourdough bread. The composition analysis, antioxidant capacity, and storage stability of the black rice sourdough bread with different proportions of black rice powder (0–60%) and fermented for different lengths of time (0–48 h) were discussed. The results showed that the black rice sourdough had the maximum lactic acid bacteria count (9 Log CFU/g) during 12 to 36 h of fermentation. The titratable acid, lactic acid, and acetic acid yields increased with the fermentation time and the proportion of black rice powder. The total anthocyanin content and antioxidant capacity increased with the fermentation time. The hardness and chewiness of the black rice sourdough bread were positively correlated with the black rice sourdough powder content and increased with storage time. In addition, the growth of fungi was significantly slowed as the additional level of black rice sourdough powder increased.

## 1. Introduction

Grain is one of the optimum matrices for developing healthy food. It contains abundant carbohydrates, proteins, vitamins, mineral substances, and dietary fibers. Moreover, cereal grain contains nondigestible carbohydrates that have beneficial effects in stimulating the growth of probiotics in the colon, consequently stabilizing intestinal flora [1]. Paddy rice (*Oryza sativa* L.) is considered safe (GRAS) and a staple food in many Asian countries. Several rice varieties have pigments, including black, red, and brown [2]. Black rice has abundant phenolic compounds, including anthocyanin, phenolic acid, flavonoids, γ-oryzanol [2,3,4], and dietary fibers [5]. Many studies have indicated that black rice can prevent dysfunction of the cardiovascular and digestive systems, aging, allergies, atherosclerosis, diabetes, oxidation, and cancers [2,4,6,7].

The lactic acid bacteria flora comprises 40 genera, with about 380 species of bacteria. The most familiar lactobacilli such as *Lactobacillus* and *Bifidobacterium* [8]. Kim et al. (2019) [9] indicated that lactic acid bacteria often exist in plants, cereals, milk products, animal intestinal tracts, and fermented food. In fermentation, lactic acid bacteria assimilate and use carbohydrates to produce various organic acids, including lactic acid and acetic acid, which can improve fermented food’s flavor and texture and benefit human health. *L. brevis* is a heterofermentative lactic bacteria that can catabolize carbohydrates, such as glucose, maltose, fructose, and xylose. Research shows that *L. brevis* can adjust intestinal microbiota [9,10,11], resist oxidation [12], inhibit colitis [13,14], have anticancer effects [15], and produce γ-aminobutyric acid (GABA) [16].

Acidification of the dough is a lactic acid bacterial activity that varies among sourdough strains and may affect the physicochemical changes throughout bread shelf-life. Sourdough improves various functions in bread or baked goods, including enhancing flavor [17,18,19], improving texture [20,21,22], extending shelf life [23], increasing mineral bioavailability [24,25], relieving celiac disease symptoms [20,26], and increasing the content of bioactive compounds [27].

According to different production methods, sourdough can be divided into Type I, Type II, and Type III sourdoughs. The microbial ecosystem of Type I sourdough is likely influenced by bread composition changes, storage temperature changes, processing environments, and manual operation [28]. Type II sourdough is industrially improved to allow faster production on a larger scale while remaining under control [18]. Type III sourdough is the flour obtained by drying Type II sourdough, which allows the lactic acid bacteria to be stored for longer for convenience [29,30]. This study explored the fermentation of black rice sourdough by lactic acid bacteria and evaluated changes in growth conditions, total phenols, anthocyanin, fermentation quotient, and antioxidant capacity of the lactic acid bacteria. The study identified optimal fermentation conditions, produced Type III black rice sourdough powder, and discusses the optimum storage temperature. Type III black rice sourdough powder was also made into black rice sourdough bread and tested for storage properties, including physical characteristics and microbial control variation. The results of the study can serve as a reference for producing sourdough bread, improving production convenience and reducing the risk of contamination. Moreover, this study helps create unique bread flavors with improved texture, antioxidant efficacy, and prolonged storage.

## 2. Materials and Methods

### 2.1. Materials, Microorganism, and Cultivation

This study used a lactic acid bacterium, *L. brevis* LUC 247, derived from Dr. Ying-Chen Lu’s strain bank at Chiayi University. We thawed the *L. brevis* LUC 247 strain from the freezer and added 0.1 mL to 0.9 mL of fresh deMan Rogosa and Sharpe (MRS) agar culture medium for activation. It was incubated at 37 °C for 24 h. Then, we added the activated bacterial solution to a fresh MRS culture medium in a volume nine times larger than the bacterial solution. This was incubated at 37 °C for 24 h for the second activation and expansion. Then, we diluted the bacterial suspension sequentially by the 10-fold dilution method until the bacterial count reached 9 log CFU/mL.

The purity of the purchased reagents and chromatography standards from Sigma-Aldrich Chemical Co. (St. Louis, MI, USA) was not lower than 98%.

### 2.2. Methods

#### 2.2.1. Effect of Black Rice on Sourdough Characteristics

To prepare the black rice sourdough, a bacterial suspension was combined with varying proportions of bread flour and black rice powder (Table 1). The resulting mixtures were then fermented at 37 °C for 48 h, with samples taken at different intervals (0, 4, 8, 12, 24, 36, and 48 h) to analyze total bacterial cell count, pH, total acid, organic acids, antioxidant activities, and more. Once the optimal fermentation conditions were achieved, the black rice sourdough was freeze-dried for 48 h. Then, it was pulverized to create Type III sourdough flour and studied for its properties during storage.

#### 2.2.2. Determination of Lactic Acid Bacterial Cell Count

The lactic acid bacterial cell count was performed after the modification by referring to the method of Siepmann et al. (2019) [18]. In brief, the sourdough (0.1 g) was mixed with 0.9 mL of 0.85% sterile saline solution. The cell suspension was serially diluted with the identical diluent to make a 10-fold dilution. Then, 0.1 mL of the diluted cell suspension was spread on MRS agar. The plate was incubated at 37 °C for 24 to 48 h. The viable cells were expressed as Log colony-forming units (CFU) per gram.

#### 2.2.3. Determination of pH Value and Titratable Acid Content

The pH value determination was modified according to Coda et al. (2010) [31]. A desktop pH meter (MP220, Metter Toledo, UK, Switzerland) was calibrated with calibration solutions with pH values of 4.01 and 7.00. Next, 5 g of sourdough was mixed with 45 mL of distilled water, and we measured the pH value.

The determination of titratable acid was modified on the basis of the work of Rühmkorf et al. (2012) [32]. Five grams of sourdough were mixed with 45 mL of distilled water and placed in a triangular flask. Next, 0.2 mL of 1% phenolphthalein indicator was added. The mixture was titrated with 0.1 N NaOH solution until the solution initially turned pink and did not fade within 0.5 min as the endpoint (pH = 8.1~8.3). The consumed volume of the NaOH solution was then recorded upon reaching the endpoint of the titration, and the titration acid content was calculated.

#### 2.2.4. Determination of Organic Acid Content and Fermentation Quotient

The determination was performed after the modification by referring to the method of Clément et al. (2018) [22]. In brief, 1 g of sourdough was diluted in 19 mL of deionized (DI) water to achieve a twentyfold dilution. The mixture was then shaken for 1 min and centrifuged at 12,000 rpm for 20 min. The supernatant was filtered using a 0.22 μm PVDF filter membrane and analyzed using high-performance liquid chromatography (HPLC) (SCL-10ADVP, Shimadzu, Kyoto, Japan). The analytical column was an Athena C18-WP (4.6 × 260 mm) (ANPEL, Shanghai, China). The mobile phase consisted of a mixture of acetonitrile and 0.05 M phosphoric acid solution (5:95 *v*/*v*), with a flow velocity of 0.6 mL/min and a column temperature of 30 °C. The chromatogram was recorded at 318 nm. The fermentation quotient was calculated as the molar concentration ratio of lactic acid to acetic acid.

#### 2.2.5. Antioxidant Capacity

DPPH free radical scavenging activity: The DPPH free radical scavenging activity was measured by Chiang et al. (2022) [33]. A total of 0.4 mL black rice sourdough extraction liquid was thoroughly mixed with 1 mL of a 0.2 mM DPPH solution. The mixture was kept still in a dark place for 50 min. The absorbance value was determined by a spectrophotometer with a 517 nm wavelength. A lower absorbance value represented a stronger ability to scavenge DPPH radicals. Vitamin C at different concentrations (0–50 μg/mL) was prepared as standards and the standard curve. The measured absorbance value of the sample was substituted into the standard curve to obtain the DPPH radical scavenging ability of the sample equivalent to that of Vitamin C. The unit was represented by the amount of Vitamin C in micrograms per gram of sourdough.

Reducing power: The reducing power was measured and modified on the basis of Chiang et al. (2022) [33]. Specifically, 0.2 mL of black rice sourdough extraction liquid was thoroughly mixed with 0.2 mL of a 0.2 M sodium phosphate buffer solution and 0.2 mL of a 1% K_3_Fe(CN)_6_ solution. The mixture was placed in a 50 °C water bath for 20 min, cooled on ice for five min, thoroughly mixed with 0.2 mL of a 10% TCA solution, and centrifuged by a high-speed centrifuge at 5000 rpm for 10 min. Then, 0.4 mL of the supernatant was thoroughly mixed with 0.4 mL DI water and 0.08 mL of a 0.1% FeCl_3_‧6H_2_ O solution and kept still in a dark place for 10 min. Finally, the absorbance value was determined by a spectrophotometer with a wavelength of 700 nm. A higher absorbance value represented stronger reducing power. Vitamin C at different concentrations (0–100 μg/mL) was prepared as standards and the standard curve. The measured absorbance value of the sample was substituted into the standard curve to obtain the reducing power of the sample equivalent to that of Vitamin C. The unit was represented by the amount of Vitamin C in micrograms per gram of sourdough.

#### 2.2.6. Determination of Total Polyphenol Content and Total Anthocyanin Content

Total polyphenol content: The method of Chiang et al. (2022) [33] was modified. In brief, 0.025 mL of black rice sourdough extraction liquid was thoroughly mixed with 1 mL of 2% Na_2_CO_3_ and kept still for two min. The mixture was thoroughly mixed with 0.25 mL of a 50% Folin–Ciocalteu reagent, kept in a dark place for 30 min, and determined by a spectrophotometer with a 750 nm wavelength. The standard curve was used for the gallic acid (0–1 mg/mL). The measured absorbance value of the sample was substituted into the standard curve to obtain the gallic acid content in the total polyphenols. The unit was represented by the amount of GAE in milligrams per gram of sourdough.

Total anthocyanin: A modification was performed according to the method of Reagents et al. (2005) [34], namely, 0.2 mL black rice sourdough extraction liquid was mixed with 0.8 mL of pH 1.0 and pH 4.5 buffer solutions and reacted in a dark room for 20 min. The measurements were performed with wavelengths of 520 nm and 700 nm, substituted into the following equation:$$\mathrm{Anthocyanin}\;\mathrm{pigment}\;(\mathrm{cyanidin}-3-\mathrm{glucoside}\;\mathrm{equivalents},\;\mathrm{mg}/L)=\frac{A\;\times \;\mathrm{MW}\;\times \;\mathrm{DF}\;\times 10^{3}}{\varepsilon \;\times \;1}$$

A = (A_520nm_ − A_700nm_) pH 1.0 − (A_520nm_ − A_700nm_) pH 4.5;

MW = molecular weight of cyanidin-3-glucoside (MW = 449.2 g/mol);

DF = dilution factor;

1 = light path length of 1 cm;

ε = 26,900, the molar extinction coefficient of cyaniding-3-glucoside; 

10^3^ = factor converted from g into mg.

#### 2.2.7. Volatile Compound Analysis

Dynamic headspace extraction (DHS) was used for this experiment. A heater was used to make sourdough generate volatile compounds. Helium gas was blown onto the sample at an adsorbent temperature during an adsorption time, and the volatile compounds were adsorbed by two sampling adsorption tubes (TenaxTA and CarbopackB/CarbopackX, GERSTEL, Mülheim and der Ruhr, Germany). Thermal desorption was performed, and a cooled injection system was used for concentrating the sample and split stream sampling. Finally, gas chromatography (Agilent, Santa Clara, CA, USA) was used for separation, and a mass spectrometer was used for analysis.

#### 2.2.8. Color Analysis

Hunter L, a, and b values for the color of Type III black rice sourdough flour were determined by a colorimeter (NE4000, Nippon Denshoku, Tokyo, Japan). The comprehensive color difference was represented by △E*, expressed as follows:△E* = [(△L*)2 + (△a*)2 + (△b*)2]^½^

#### 2.2.9. Determination of Water Activity and Moisture

The sample was measured with a water activity meter (AQUALAB, Pullman, WA, USA), and the error value was 0.01 Aw (Reale et al., 2019 [30]). A moisture analyzer (MOC63u, Shimadzu, Kyoto, Japan) was used for analysis, during which 3 g of the sample was placed on the moisture measuring tray. After being heated by 105 °C infrared rays, the sample weight was detected at a frequency of every 0.2 s until the moisture had evaporated and the sample maintained a constant weight.

#### 2.2.10. Production of Black Rice Sourdough Bread

The bread formula was modified using different proportions of bread flour and black rice sourdough powder, as Tafti et al. (2013) [35] outlined. Four different proportions were used, including BB0 (no black rice sourdough powder), BB10 (10% black rice sourdough powder), BB20 (20% black rice sourdough powder), and BB30 (30% black rice sourdough powder). The ingredients used for the bread included bread flour, black rice sourdough powder, distilled water, yeast powder, salt, granulated sugar, milk powder, and salt-free cream. An intelligent automatic bread machine was used for production, which included mixing the ingredients at low speed for 3 min, stirring at medium speed for 10 min, keeping the mixture still for 15 min, stirring again at medium speed for 10 min, fermenting at 80% humidity for 27 min, reshaping slowly for 5 min, referencing at 80% humidity for 25 min, and finally baking at 150 °C for 1 h.

#### 2.2.11. Analysis of Physical Properties of Black Rice Sourdough Bread

The method of Kiumarsi et al. (2019) [36] was modified. Briefly, the analysis was performed by the texture profile analysis (TPA) method using a physical property analyzer (TA.XTPlus, Stable Micro Systems, Godalming, England). The entire loaf was cut into a 2.5 cm thin slice, which was tested by a P50 probe. The amount of compression was 40%, the probe speed; the test speed was 5 mm/s; and the probe was pressed down twice for a measurement to obtain the bread’s hardness, springiness, chewiness, cohesiveness, and resilience.

#### 2.2.12. Test for Storage Property of Black Rice Sourdough Bread

The completed black rice sourdough bread was cooled for 30 min. The bread was then cut into 2.5 cm slices and stored in a zipper bag. A small opening remained for exchanging air with the outside. The slices were stored at 25 °C and observed every day. The experiment stopped when fungi started growing on the surfaces of the slices. The fungal growth area was measured and calculated using ImageJ image processing software [37].

#### 2.2.13. Statistical Analysis

All experiments were performed three times. SPSS statistical analysis software (IBM Statistics 19) was used for analysis. The differences among various groups were verified by one-way ANOVA, and significant differences were analyzed by Duncan’s multiple range test, in which *p* < 0.05 represented a significant difference.

## 3. Results and Discussion

### 3.1. Effect of Black Rice Powder Content on the Growth of L. brevis

On the basis of the growth curve of the lactic acid bacterial count shown in Figure 1A, the bacterial counts of all groups increased rapidly during the first 12 h of fermentation. BS30 had the highest bacterial count among all groups, reaching 9.22 log CFU/g. The stationary phase of the bacterial count was observed between 12 and 36 h of fermentation, during which the nutrients in the black rice sourdough were almost depleted, leading to a dynamic equilibrium between living and dead bacterial counts. The bacterial count of BS60 started to decline after 24 h of fermentation. The bacterial counts of BS0, BS15, BS30, BS45, and BS60 were 8.92, 8.18, 8.64, 7.99, and 7.84 CFU/g, respectively, at the end of 48 h of fermentation, indicating that the nutrients in the black rice sourdough were no longer sufficient to support the growth of the strains, causing cell death. Therefore, the optimal fermentation time for the black rice sourdough was between 12 and 36 h, during which the maximum bacterial count was observed. The dough’s nutrient content and organic acid accumulation were critical factors in controlling the bacterial count. The lactic acid bacterial cell count was maintained at the peak point by making bread using the black rice sourdough in this fermentation interval. The strain viability was better, and more nutrient metabolites were able to be generated.

Starch degradation is an essential part of the sourdough fermentation process. The co-activation depends on the activity of amylase in the flour and the glucosidase in the species. After degradation, multiple fermentable sugars are generated to provide nutrients required by sourdough’s lactic acid bacteria and yeast [18,38]. Black rice bran contains carbohydrates, proteins, oil, fat, and micronutrients. These nutrients can enhance lactic acid bacteria’s growth and survival abilities [39].

### 3.2. pH and Titratable Acid Content of Black Rice Sourdough

The pH variation of the black rice sourdough with different proportions of black rice powder is shown in Figure 1B. The pH values of all groups decreased rapidly from 0 to 24 h of fermentation, and the pH value decreased slowly after 24 h of fermentation. The pH value ranged from 3.77 to 3.83 after 48 h of fermentation.

The variation of the titratable acid content in the black rice sourdough is shown in Figure 1C. The titratable acid of all groups increased at the highest rate after 8 to 36 h of fermentation, possibly because the lactic acid bacteria were in a logarithmic growth phase. During that time, the strains multiplied, the bacterial count remained high, and a large amount of organic acid was generated. Hence, the group with a higher proportion of black rice powder had a higher titratable acid content. The titratable acid increased slowly from 36 to 48 h because the lactic acid bacteria began to decline and die as the nutrition was exhausted. The pH value was negatively correlated with the titratable acid curve. Probiotics consume fermentable sugars as carbon sources for growth and generate organic acids. The organic acid enhances the protease and amylase activity in the flour, induces protein and starch degradation, and provides more nutrient substances for the growth of lactic acid bacteria [38]. The higher the additional level of black rice powder, the greater the yield of organic acid, as black rice contains a great amount of fermentable carbohydrates.

The pH value is a critical factor in the survival of probiotic lactic acid bacteria, and acid accumulation during fermentation may be the reason for the decrease in lactic acid bacterial counts. Pereira et al. (2017) [40] indicated that a pH above 4.0 would not influence lactic acid bacteria, but a pH below 4.0 might inhibit their existence. Therefore, decrease in lactic acid bacterial cell count might have been caused by inhibition due to the excessive accumulation of organic acid. The lactic acid bacterial count of the BS60 decreased after 36 h and might have been inhibited earlier than the other groups due to its higher titratable acidity.

### 3.3. Organic Acid Content and Fermentation Quotient of Black Rice Sourdough

Organic acid provides an antibacterial function and flavor for sourdough. The lactic acid bacteria used in this study were heterofermentative lactic bacteria. Acetic acid was produced in addition to lactic acid; therefore, this study simultaneously discussed the fermentation-induced changes in the dough’s lactic acid and acetic acid contents.

Figure 1D,E shows the changes in the black rice sourdough’s lactic acid and acetic acid contents using different proportions of black rice powder fermented for 48 h. The lactic acid and acetic acid contents significantly increased with fermentation time during the fermentation process. The higher the addition level of black rice powder was, the higher the lactic acid and acetic acid contents. BS60 had the highest lactic and acetic acid contents after 48 h of fermentation, at 7.30 and 5.71 times for initial lactic acid and acetic acid contents, respectively.

As rice germ has more sucrose than wheat, heterofermentative lactic bacteria can generate more lactic acid and acetic acid when the germ is added to sourdough [33]. Therefore, adding black rice powder can increase lactic acid and acetic acid yields. Rehman et al. (2006) [41] found that *L. brevis* inhibits the growth of endogenous yeast in sourdough, causing ethanol to decrease and the acetic acid yield to increase. Homofermentative lactic bacteria produce 2 moles of lactic acid by consuming 1 mole of glucose, but heterofermentative lactic bacteria only obtain 1 mole of lactic acid by consuming 1 mole of glucose. Therefore, the fermentation quotient (0.58–1.14) of this study was lower than the sourdough fermentation quotient (1–17) of Ravyts and De Vuyst (2011) [42].

### 3.4. Anthocyanin and Total Phenol Contents and Antioxidant Capacity of Black Rice Sourdough

The anthocyanin and total phenol content in the black rice sourdough using different proportions of black rice powder is shown in Table 2. The anthocyanin content decreased and then increased significantly during 0 to 8 h of fermentation. A higher level of black rice powder resulted in the group’s total anthocyanin content. The total phenol content increased with the fermentation time and was positively correlated with the level of black rice powder. BS45 and BS60 reached the maximum value after 36 h and decreased after 48 h. BS45 had the highest total phenol content after 36 h of fermentation (2.29 mg GAE/g sourdough), 3.36 times the initial total phenol content. Table 2 also shows the antioxidant capacity of black rice sourdough. The DPPH radical scavenging ability and reducing power increased with fermentation time and positively correlated with the added black rice powder. The antioxidant capacity of all groups reached the maximum value after 36 h of fermentation and then decreased after 48 h of fermentation. BS45 had the strongest DPPH radical scavenging ability (1017.57 Vit. C equivalent; μg/g sourdough) and reducing power (668.10 Vit. C equivalent: μg/g sourdough) after 36 h of fermentation, which was 4.77 and 2.70 times the initial fermentation, respectively.

Lin and Chou (2009) [43] found that the extractable anthocyanin content increases after black beans are fermented by *Aspergillus awamori*. Because β-glucosidase catalyzes the hydrolysis of the covalent bond between the phenolic compounds and the cell wall matrix, anthocyanin is released from plant tissues. Phenolic compounds, especially anthocyanin, are considered natural antioxidants and one of black rice bran’s most abundant phytochemical materials [2]. Lactic acid bacteria use polysaccharides to generate phenolic esterase and carbohydrase, which can hydrolyze the ester bond between phenols and the plant cell wall. As a result, soluble conjugate phenols are released in free form, and the total phenol content in the bran and antioxidant activity increases [44,45]. As shown in Table 2, the DPPH radical scavenging ability and reducing power had similar variation trends. Therefore, the DPPH radical scavenging ability and reducing power of the black rice sourdough increased with the level of black rice powder.

The highest lactic acid bacterial count was observed between 12 and 36 h of fermentation. The maximum levels of lactic acid and acetic acid were reached after 48 h of fermentation. The content of total anthocyanins and phenols had a positive correlation with the amount of black rice powder added and peaked after 36 h of fermentation. The antioxidant capacity of the BS45 strain was highest after 36 h of fermentation. On the basis of these results, the optimal fermentation conditions for black rice sourdough with 45% black rice powder were identified as 36 h of fermentation.

### 3.5. Volatile Compounds of Black Rice Sourdough

A volatile compound analysis is shown in Figure 2 for the control dough (dough without lactic acid bacteria, where fermentation was solely dependent on endogenous microbes) and black rice sourdough fermented for 36 h with 45% black rice powder. 

Figure 2 compares the volatile compounds sampled by the Carbopack B/Carbopack X adsorption tube and the control group. The results show that BS45 increased from 0% to 0.03% in alkanes, 85.25% to 92.09% in carboxylic acid, 0% to 2.57% in ketone ethers, and 0.74% to 4.53% in esters, and decreased from 14.02% to 0.79% in alcohols.

The volatile compounds sampled by the Tenax TA adsorption tube were compared with the control group. The alkanes of the BS45 increased from 0.06% to 0.12%, and no alcohols were observed. The carboxylic acid decreased from 95.02% to 62.56%, the ketone ethers increased from 0% to 16.77%, and the esters increased from 4.92% to 20.55%. As shown in Figure 2, BS45 contained more ketones, ether, and esters than the control group. It often had the fragrance of flowers and fruit, nuts, and cream. Therefore, sourdough could be used as a fragrance modifier for baked food [46]. To sum up, the fermentation of black rice sourdough with 45% black rice powder for 36 h was the optimal fermentation condition. The black rice sourdough was freeze-dried and made into Type III sourdough flour and then tested at 4 °C and 25 °C for 0, 2, 4, and 6 weeks to determine its storage properties.

### 3.6. Variation of Lactic Acid Bacteria Content in Type III Black Rice Sourdough Powder at Different Storage Temperatures

The variation of the lactic acid bacterial counts of the black rice sourdough powder stored at 4 °C and 25 °C for 6 weeks is shown in Table 3. The lactic acid bacterial count decreased with storage time at the two temperatures; the lactic acid bacterial count during storage at 4 °C and 25 °C in Week 6 was 7.13 and 7.06 Log CFU/g. About 2 Log CFU/g reduced the bacterial count after freeze-drying in comparison with the black rice sourdough before freeze-drying. Choi et al. (2019) [47] indicated that freeze drying removes water from frozen food by sublimation; ice crystals are formed in this process, causing cells to be damaged and cell viability to be degraded.

### 3.7. Variation of Moisture Content and Water Activity of Type III Black Rice Sourdough Powder at Different Storage Temperatures

The black rice sourdough powder’s moisture content and water activity at different storage temperatures decreased with storage time. However, there was a significant difference between the two storage temperatures (Table 3). Chávez and Ledeboer (2007) [48] indicated that the water activity of food powder with probiotics should be lower than 0.25, and the moisture content should be lower than 5% to prevent cell death during storage. In this study, the initial water activity was 0.13, and the moisture content was 2.75%, meeting the conditions outlined.

### 3.8. Variation of Total Phenol Content and Anthocyanin Contents in Type III Black Rice Sourdough Powder at Different Storage Temperatures

As shown in Table 3, the total phenol content at 25 °C decreased significantly after six weeks’ storage at 4°C but did not differ significantly after six weeks’ storage at 4 °C. The metabolic reactions of grains may be influenced by temperature, relative humidity, and gas composition during long-term storage. After four months of storage at 20, 30, and 40 °C, black and red rice’s total phenolic compound contents decreased [4]. Yang et al. (2017) [49] found that the total phenol content decreased significantly with an increase in storage time and temperature, which matched the findings of this paper.

The variation of anthocyanin content (Table 3) showed that at the two storage temperatures, the decrease in anthocyanin content reached the maximum of 42% at week 2 of storage, after which it stabilized. The stability of the anthocyanin in the black rice grains was directly correlated with storage time, light, temperature, and oxygen concentration. The reduction of the total anthocyanin content, cornflower-3-glucoside, and methyl-3-glucoside during storage was related to grains’ oxidation and enzyme reaction during storage [4]. In this study, the freeze-dried black rice sourdough powder was exposed to a high oxygen concentration before packaging, and some oxygen remained after packaging. As a result, enzymes were prone to react, and the anthocyanin was degraded, leading to a significant decrease in the anthocyanin at the two storage temperatures in weeks 0–2. Oxygen was consumed gradually with the storage time, and as a low oxygen environment was presented in the sealed package, the stability of the anthocyanin was able to be maintained.

### 3.9. Color Change of Type III Black Rice Sourdough Powder at Different Storage Temperatures

During storage at 4 °C and 25 °C, the L* value and a* value significantly decreased as the storage time increased, and the b* value increased with the storage time. △E* also significantly increased with storage time (Table 3). The △E* in Week 6 was 2.33 and 1.19, compared with the initial storage. Black rice contains a significant amount of anthocyanin, a flavonoid responsible for giving flowers, vegetables, fruits, and grains their distinctive blue, red, purple, and orange hues. The color of anthocyanin varies depending on the pH level [50]. However, during storage, anthocyanin can be degraded by oxygen and residual enzyme activity, leading to color changes in the rice [51].

### 3.10. Texture Analysis of Black Rice Sourdough Bread during Storage

According to Figure 3, black rice sourdough bread made with different proportions of black rice sourdough powder changes its hardness (A), springiness (B), cohesiveness (C), resilience (D), and chewiness (E) after four days of storage. A positive correlation was found between storage time and hardness, and chewiness. The springiness, cohesiveness, and resilience decreased as the storage time increased. Gluten-free bread often ages quickly because its principal raw material is pure starch, causing it to age faster than wheat bread. After nine days’ storage, the hardness of wheat bread and bread made with rice husk is higher than that during the initial storage, and the aging rate of bread made with rice husk is significantly higher than that of wheat bread [52].

### 3.11. Analysis of Fungal Growth of Black Rice Sourdough Bread during Storage

The point when fungal growth begins is very important for bread; it is related to moisture content, organic acid, and the existence of bacteriostatic materials. Delaying the beginning of fungal growth can prolong the storage life of bread to enhance food safety for consumers.

Figure 4 and Figure 5 show the fungal growth image and fungal growth area of the black rice sourdough bread with different proportions of black rice sourdough powder during storage. As shown in Figure 4, BB0 had fungal growth on day 2 of storage, BB10 and BB20 had fungal growth on day 3, and BB30 had fungal growth on day 4. On day 4 of storage, the fungal growth areas were 36.87, 9.53, 8.79, and 0.24%, respectively.

Fungi mainly induce bread rot. It is known that lactic acid bacteria can generate organic acid to reduce pH and prevent miscellaneous bacteria from growing. In addition, bacteriostatic substances can be generated, such as phenyl carboxylic acid and its derivatives, cyclic dipeptides, and antifungal peptides. Axel et al. (2016) [38] indicated that *L. brevis* R2Δ could be used as a bioprotective bacterium source as its metabolites have antifungal activity. As *L. brevis* generates antibiotic substances in the fermentation process, the beginning of fungal growth can be postponed by increasing the proportion of black rice sourdough. In this study, BB30 had fungal growth two days later than BB0.

## 4. Conclusions

This study used *L. brevis* LUC 247 to ferment black rice sourdough, which was then freeze-dried to make Type III sourdough flour. Black rice sourdough bread was made according to different proportions, and the optimal production conditions were discussed. The use of Type III sourdoughs improved the overall characteristics of bread. In addition, a specific lactic acid bacteria strain, *L. brevis* LUC 247, in sourdough breadmaking may delay hardness, chewiness, staling, and mold growth. Moreover, it enhances the antioxidant activity and bioactive ingredients of sourdough. The study’s findings suggest that bread made with *L. brevis* LUC 247 had superior storage capabilities, flavor, texture, and bioactive compound content. The optimal production conditions identified for making black rice sourdough bread in this study could be a valuable reference for future production.

## Figures and Tables

**Figure 1 foods-12-01389-f001:**
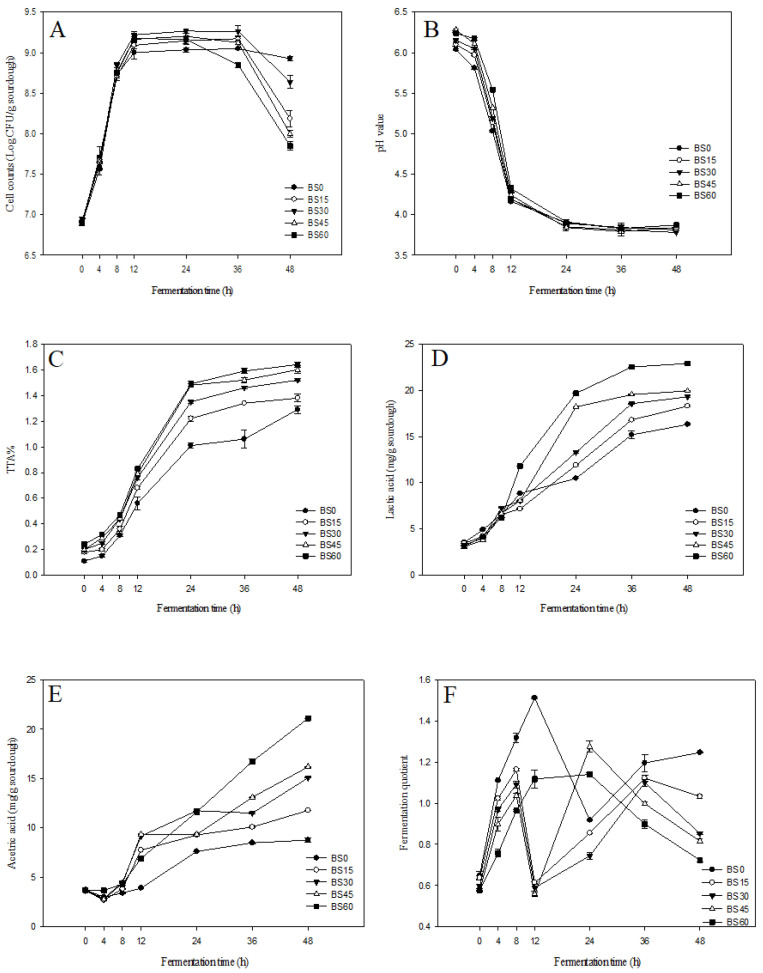
Lactic acid bacteria content (**A**), pH value (**B**), titrate acidic quantity (**C**), lactic acid concentration (**D**), acetic acid concentration (**E**), and fermentation quotient (**F**) of different black rice powder proportions of black rice sourdough fermented by *L. brevis* LUC 247. BS0: without black rice powder; BS15: with 15% black rice powder; BS30: with 30% black rice powder; BS45: with 45% black rice powder; and BS60: with 60% black rice powder.

**Figure 2 foods-12-01389-f002:**
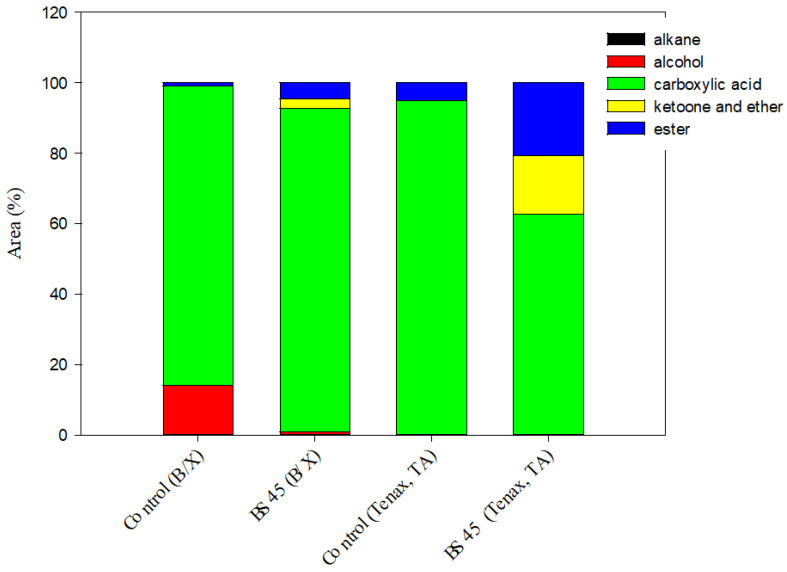
Volatile compound composition of control dough and black rice sourdough with added 45% black rice powder fermented for 36 h. B/X: volatile compounds were sampled with carbopack B/carbopack X adsorption tubes; TA: volatile compounds were sampled with tenax TA adsorption tubes.

**Figure 3 foods-12-01389-f003:**
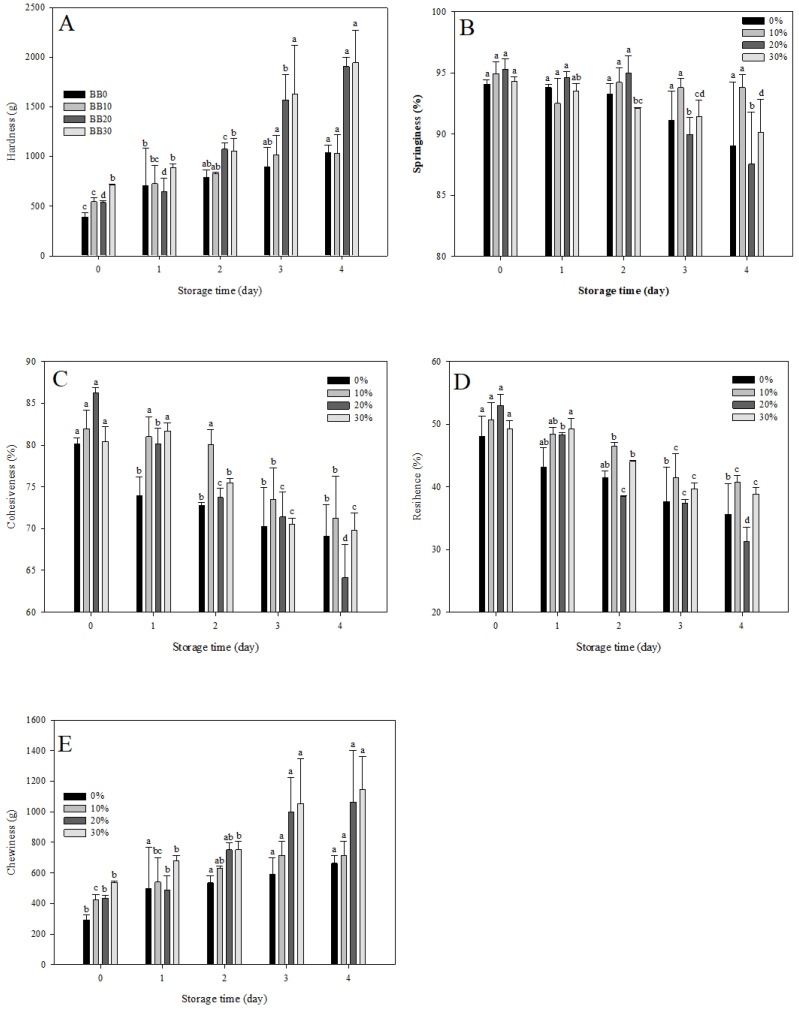
(**A**) Hardness, (**B**) springiness, (**C**) cohesiveness, (**D**) resilience, and (**E**) chewiness of different black rice sourdough powder proportions of black rice sourdough bread during storage. BB0: without black rice sourdough powder; BB10: with 10% black rice sourdough powder; BB20: with 20% black rice sourdough powder; and BB30: with 30% black rice sourdough powder. a–d: Data with the identical letter in the same column are not significantly different (*p* < 0.05).

**Figure 4 foods-12-01389-f004:**
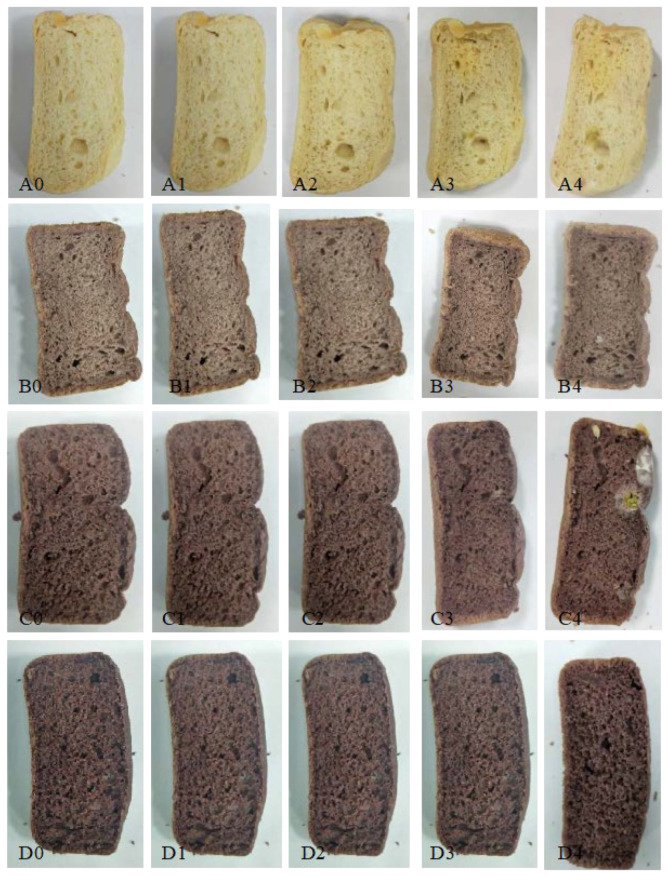
Mold growth of different black rice sourdough powder proportions of black rice sourdough bread. Codes A, B, C, and D represent BB0, BB10, BB20, and BB30 groups, respectively, and codes 0, 1, 2, 3, and 4 represent storage days 0, 1, 2, 3, and 4, respectively.

**Figure 5 foods-12-01389-f005:**
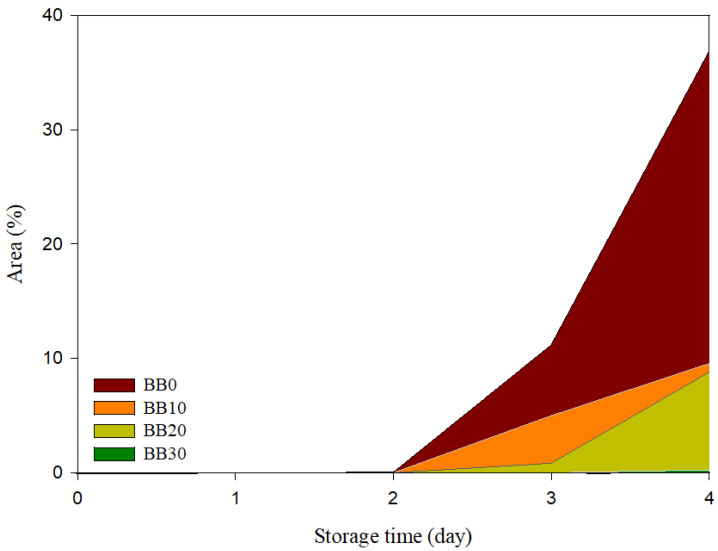
The mold growth area of black rice sourdough bread with different proportions of black rice sourdough powder was added. BB0: without black rice sourdough powder; BB10: with 10% black rice sourdough powder; BB20: with 20% black rice sourdough powder; and BB30: with 30% black rice sourdough powder.

**Table 1 foods-12-01389-t001:** Formula of black rice sourdough.

Materials	Black Rice Sourdough				
	BS0 *	BS15	BS30	BS45	BS60
Black rice powder (g)	0	30	60	90	120
Bread flour (g)	200	170	140	110	80
Sterilized distilled water (mL)	200	200	200	200	200
Bacterial suspension (mL) **	4	4	4	4	4

* BS0 refers to black rice sourdough bread dough with 0% black rice flour added; BS15 refers to black rice sourdough bread dough with 15% black rice flour added; BS30 refers to black rice sourdough bread dough with 30% black rice flour added; BS45 refers to black rice sourdough bread dough with 45% black rice flour added; BS60 refers to black rice sourdough bread dough with 60% black rice flour added. ** lactic acid bacterial cell counts of approximately 9 Log CFU/mL.

**Table 2 foods-12-01389-t002:** Contents of anthocyanins, total phenols, and the antioxidant capacity of black rice sourdough with different proportions of black rice powder fermented for 48 h.

	Sample	Fermentation Time (h)						
		0	4	8	12	24	36	48
Anthocyanin (μg/g sourdough)	BS0	N.D. *	N.D.	N.D.	N.D.	N.D.	N.D.	N.D.
	BS15	4.23 ± 0.19 ^dF^	2.23 ± 0.39 ^dG^	8.13 ± 0.39 ^dE^	16.70 ± 2.03 ^dD^	28.72 ± 3.22 ^dC^	31.73 ± 0.33 ^dB^	42.40 ± 0.26 ^dA^
	BS30	15.92 ± 0.13 ^cC^	8.91 ± 0.20 ^cD^	16.36 ± 0.33 ^cC^	37.85 ± 0.70 ^cB^	54.44 ± 0.88 ^cA^	56.44 ± 0.58 ^cA^	56.22 ± 0.39 ^cA^
	BS45	36.63 ± 0.15 ^bC^	25.27 ± 0.21 ^bD^	23.60 ± 0.51 ^bD^	52.66 ± 0.91 ^bB^	75.41 ± 1.20 ^bA^	78.15 ± 1.46 ^bA^	71.58 ± 0.51 ^bA^
	BS60	63.46 ± 1.77 ^aB^	48.76 ± 1.20 ^aC^	52.99 ± 0.70 ^aC^	62.34 ± 0.77 ^aB^	86.83 ± 0.0 ^aA^	86.28 ± 0.29 ^aA^	81.94 ± 1.17 ^aA^
Total phenolic (gallic acid equivalent;mg/g sourdough)	BS0	0.55 ± 0.01 ^bD^	0.65 ± 0.01 ^bD^	1.14 ± 0.00 ^cC^	1.29 ± 0.01 ^cC^	1.40 ± 0.01 ^dB^	1.70 ± 0.06 ^bA^	1.78 ± 0.07 ^bA^
	BS15	0.54 ± 0.02 ^bD^	0.55 ± 0.01 ^cD^	1.53 ± 0.01 ^bC^	1.55 ± 0.05 ^bC^	1.59 ± 0.03 ^cC^	1.79 ± 0.02 ^bB^	2.01 ± 0.02 ^aA^
	BS30	0.60 ± 0.02 ^aD^	0.58 ± 0.01 ^cD^	1.29 ± 0.02 ^cC^	1.74 ± 0.03 ^aB^	1.87 ± 0.08 ^bB^	2.11 ± 0.03 ^aA^	2.17 ± 0.02 ^aA^
	BS45	0.68 ± 0.02 ^aD^	0.71 ± 0.02 ^aD^	0.88 ± 0.01 ^dC^	1.63 ± 0.03 ^bB^	2.00 ± 0.01 ^aA^	2.29 ± 0.05 ^aA^	2.17 ± 0.01 ^aa^
	BS60	0.68 ± 0.00 ^aC^	0.78 ± 0.01 ^aC^	1.64 ± 0.02 ^aB^	1.69 ± 0.03 ^aB^	2.00 ± 0.01 ^aA^	2.15 ± 0.05 ^aA^	2.06 ± 0.03 ^aA^
DPPH RSA (Vit. C equivalent; μg/g sourdough)	BS0	48.25 ± 3.29 ^dD^	53.04 ± 6.31 ^eD^	75.42 ± 0.99 ^eC^	75.55 ± 2.67 ^dC^	93.45 ± 3.68 ^dB^	104.27 ± 2.88 ^dA^	73.32 ± 1.72 ^dC^
	BS15	123.73 ± 1.45 ^cD^	137.68 ± 5.09 ^dD^	178.76 ± 3.97 ^dC^	334.39 ± 2.06 ^cB^	494.17 ± 5.02 ^cA^	521.94 ± 7.77 ^cA^	504.58 ± 3.02 ^cA^
	BS30	182.86 ± 2.02 ^bC^	217.41 ± 4.90 ^cB^	237.68 ± 2.74 ^cB^	657.64 ± 5.97 ^bA^	652.53 ± 8.47 ^bA^	675.91 ± 3.59 ^bA^	671.16 ± 7.40 ^bA^
	BS45	193.53 ± 3.63 ^bE^	313.35 ± 3.02 ^bD^	378.01 ± 4.66 ^bD^	726.50 ± 9.21 ^aC^	954.37 ± 22.28 ^aB^	998.57 ± 7.30 ^aA^	923.32 ± 6.23 ^aB^
	BS60	380.05 ± 5.27 ^aE^	468.05 ± 8.07 ^aD^	573.81 ± 3.84 ^aC^	734.55 ± 8.63 ^aB^	965.15 ± 18.12 ^aA^	995.65 ± 19.24 ^aA^	905.42 ± 12.40 ^aA^
Reducing power (Vit. C equivalent; μg/g sourdough)	BS0	59.65 ± 1.26 ^eC^	60.83 ± 1.28 ^eC^	58.03 ± 1.28 ^dC^	63.05 ± 0.75 ^dC^	78.03 ± 0.57 ^eB^	76.88 ± 1.36 ^dB^	90.63 ± 2.47 ^dA^
	BS15	120.28 ± 0.51 ^dC^	115.18 ± 2.04 ^dC^	143.97 ± 0.51 ^cB^	308.90 ± 1.89 ^cA^	345.04 ± 6.70 ^dA^	339.93 ± 1.67 ^cA^	331.87 ± 6.25 ^cA^
	BS30	189.32 ± 3.9 ^cD^	176.08 ± 4.29 ^cD^	192.17 ± 4.20 ^bD^	397.88 ± 4.22 ^bC^	479.03 ± 9.39 ^cB^	506.02 ± 2.10 ^bA^	477.05 ± 13.87 ^bB^
	BS45	254.72 ± 3.21 ^bC^	248.09 ± 3.62 ^bC^	232.27 ± 3.21 ^bC^	547.26 ± 4.54 ^aB^	589.83 ± 3.49 ^bB^	688.10 ± 5.59 ^aA^	616.03 ± 14.21 ^aB^
	BS60	330.65 ± 5.78 ^aD^	304.82 ± 4.34 ^aD^	437.38 ± 8.87 ^aC^	569.73 ± 4.19 ^aB^	653.53 ± 21.08 ^aA^	665.38 ± 1.42 ^aA^	629.17 ± 7.59 ^aA^

* N.D.: not detected. Results from three separate experiments are expressed as mean ± SD. A–G: data with identical letters in the same row are not significantly different (*p* < 0.05). a–d: data with the identical letter in the same column are not significantly different (*p* < 0.05). BS0: without black rice powder; BS15: with 15% black rice powder; BS30: with 30% black rice powder; BS45: with 45% black rice powder; and BS60: with 60% black rice powder.

**Table 3 foods-12-01389-t003:** The content of lactic acid bacteria, moisture, water activity, total phenols, total anthocyanins, and color change of Type III black rice sourdough flour during storage.

Storage Temperature (°C)	Storage Time (Week)	Lactic Acid Bacteria (Log CFU/g Sourdough Powder)	Water Activity (Aw)	Moisture (%)	Total Phenolic (mg/g Sourdough Powder)	Anthocyanin (μg/g Sourdough Powder)	Color			
							L*	a*	b*	ΔE*
4 °C	0	7.25 ± 0.0 ^a^	0.13 ± 0.01 ^a^	2.75 ± 0.05 ^a^	6.75 ± 0.26 ^a^	143.72 ± 4.15 ^a^	55.81 ± 0.14 ^a^	13.71 ± 0.10 ^ab^	2.67 ± 0.06 ^c^	—
	2	7.13 ± 0.05 ^b^	0.12 ± 0.0 ^ab^	2.71 ± 0.04 ^a^	6.31 ± 0.20 ^a^	82.16 ± 0.88 ^c^	56.20 ± 0.43 ^a^	13.95 ± 0.24 ^a^	3.21 ± 0.16 ^a^	0.90 ± 0.03 ^c^
	4	7.07 ± 0.11 ^b^	0.10 ± 0.02 ^b^	2.51 ± 0.07 ^b^	6.19 ± 0.16 ^a^	89.39 ± 3.88 ^b^	54.44 ± 0.15 ^b^	13.64 ± 0.12 ^b^	2.76 ± 0.04 ^c^	1.38 ± 0.18 ^b^
	6	7.13 ± 0.12 ^b^	0.11 ± 0.01 ^b^	2.46 ± 0.12 ^b^	6.17 ± 0.19 ^a^	90.06 ± 2.15 ^b^	53.26 ± 0.35 ^c^	13.50 ± 0.04 ^c^	2.97 ± 0.07 ^b^	2.33 ± 0.11 ^a^
25 °C	0	7.25 ± 0.0 ^a^	0.13 ± 0.01 ^a^	2.75 ± 0.05 ^a^	6.75 ± 0.26 ^a^	143.72 ± 4.15 ^a^	55.81 ± 0.14 ^ab^	13.71 ± 0.10 ^ab^	2.67 ± 0.06 ^b^	—
	2	6.98 ± 0.03 ^b^	0.10 ± 0.01 ^b^	2.44 ± 0.02 ^b^	6.12 ± 0.36 ^ab^	83.61 ± 1.26 ^b^	55.47 ± 0.42 ^b^	13.90 ± 0.32 ^a^	3.22 ± 0.23 ^a^	0.66 ± 0.10 ^b^
	4	6.99 ± 0.15 ^b^	0.09 ± 0.01 ^b^	2.19 ± 0.16 ^b^	5.86 ± 0.04 ^b^	87.28 ± 2.37 ^b^	56.22 ± 0.41 ^a^	13.56 ± 0.05 ^ab^	2.64 ± 0.05 ^b^	0.71 ± 0.04 ^b^
	6	6.95 ± 0.08 ^b^	0.10 ± 0.01 ^b^	2.33 ± 0.23 ^b^	5.80 ± 0.44 ^b^	87.06 ± 1.84 ^b^	54.50 ± 0.33 ^c^	13.44 ± 0.25 ^b^	2.80 ± 0.04 ^b^	1.19 ± 0.05 ^a^

Results from three separate experiments are expressed as mean ± SD. a–d: data with the identical letter in the same column are not significantly different (*p* < 0.05).

## Data Availability

Data are contained within the article.

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
