# Peer review of "Development and Optimization of Black Rice (Oryza sativa L.) Sourdough Fermented by Levilactobacillus brevis LUC 247 for Physicochemical Characteristics and Antioxidant Capacity"

_foods, 2023, doi:10.3390/foods12071389_

Round 1

Reviewer 1 Report (Previous Reviewer 1)

Dear Authors, 

After the evaluation, the authors lack expertise in the microbiology field. Once again, I found so many mistakes in the manuscript, particularly English writing in part of microbiology which is not only limited in methodology but also results and discussion. Previously, I commented that this manuscript needs English editing and proofreading prior to submission. In this submission, however, the authors did not properly respond to my request. The manuscript was prepared  with careless consideration and concern, especially to my comments. There are grammatical errors, incomplete sentences, and misleading sentences throughout the manuscript. 

Major comments

1. The authors still misunderstand about prebiotic and probiotic concepts. Probiotics are referred to good microorganisms mostly lactic acid bacteria that when administered in an adequate amount, confer health benefits. Prebiotics, in general, are nutrient for probiotics. Prebiotic must resist to gastrointestinal system, thus passing through the GI tract to reach large intestinal where they can be fermentable by microbiota. The authors are always confused when expressing about prebiotic and probiotic concept. For example,

Lines 32-34 the authors state that it (referring to grains) contains probiotic for enterobacteria species, …. which is not true.

2. The authors lacks basic in microbiology as I have previously mentioned but I have not seen any improvement in this manuscript. For example

Line 43, The most familiar lactobacilli are Lactobacillus, ……, and Enterococcus. However, Enterococcus is a cocci genus.

3. No information about how was L. brevis cultured? What kind of media and their composition? What was the cultured conditions? And How did the author initially set up 9 log CFU/mL? Please add this information in section 2.1 which needs to be edited to “Materials, microorganism, and cultivation”

4. Section 2.2.1, I do not understand how the authors prepare bacterial suspension for dough making (line 86-89). Please rewrite to make it more understandable.

Line 86, “The bacteria solution” Is it the bacterial culture?

Line 86, I do not understand “accounting for 1% of the total dough weight” what does it mean?

Line 88, “The supernatant was removed, and 4 mL of the bacteria solution was uniformly mixed with 200 mL of sterilized distilled water. Does it make sense? The authors removed supernatant and why was it retained 4 mL.

The supernatant was removed, then the cell pellet was resuspended with 4 mL sterile water. The resulting cell suspension was mixed with 200 mL sterile water. Correct?

Line 96, remove "For various test measurement," and add more information. What were these samples analyzed? Total bacterial cell count, pH, total acid, organic acids, and so on.

Line 98, Is it for bread making? if so, please indicate in this section.

There is no methodology for section 3.6.

5. Section 2.2.2, there are huge mistakes and misleading in this section.

Line 102, “The modification was performed …. (2019)” Does it make sense? Will the readers understand the meaning that the authors would like to indicate? Please rewrite this paragraph. We do not say “aseptic saline solution”, “smearing for spread plate technique” in terms of microbiology. The condition stated in lines 105-106 does not develop the anaerobic condition at all !!!

6. Section 2.2.3, I do not think that the authors modified method for pH measurement. This is too simple.

Line 113-116, how did the authors observed the end point of titration?

7. Section 2.2.4, again, the sentence started with “A modification” which is not suitable.

Lines 125-127 contain incomplete sentence.

8. Section 2.2.10, again, the authors started the sentence with “A modification”

 Which is not referred to methodology.

9. Section 3.1, Please avoid initiating the sentence with “Fig. shows” and check throughout the manuscript.

Please replace “bacteria count” by “bacterial cell count” and edit throughout the manuscript.

Please avoid describing growth of bacteria using “peak” it is not really scientific description.

Line 262-263, the authors misunderstand about probiotic “These nutrients can act as probiotics to …” which is not true !! Black rice bran contains various nutrients which some can act as prebiotics such as xylan and its oligosaccharides (xylooligosaccharides)

10. Section 3.2,

Line 281, Has L. brevis the strain that used in this study been claimed as probiotic? If not, please edit line 281 to “Lactic acid bacteria consume fermentable sugars”

Line 290, black rice does not contain many probiotics. It may contain some lactic acid bacteria. If this is not misleading, please provide related reference.

Please edit Figure 1, Figure 2, and Figure 3 by extending the font size of axis titles and legend as they are too small.

Please look at the attachment for other comments

Author Response

03/03/23

Prof. Dr. Arun K. Bhunia
Editor-in-Chief
Foods

Dear Professor,

Thank you for considering the Resubmit of our manuscript: Foods-2258232, entitled " Development and optimization of black rice (Oryza sativa L.) sourdough bread fermented by Levilactobacillus brevis LUC 247 for physicochemical characteristics, and antioxidant capacity", by Lai. et al. for publication in Foods. We are thankful to the referees and the Editor for pointing out some important modifications needed in the report. We believe the comments have been highly constructive and useful in restructuring the manuscript. We have thoughtfully taken into account these comments. The explanation of what we have changed in response to the reviewers’ concerns is given point by point in the following pages.

Review 1

Comments and Suggestions for Authors

Dear Authors, 

After the evaluation, the authors lack expertise in the microbiology field. Once again, I found so many mistakes in the manuscript, particularly English writing in part of microbiology which is not only limited in methodology but also results and discussion. Previously, I commented that this manuscript needs English editing and proofreading prior to submission. In this submission, however, the authors did not properly respond to my request. The manuscript was prepared with careless consideration and concern, especially to my comments. There are grammatical errors, incomplete sentences, and misleading sentences throughout the manuscript. 

  1. The reviewer’s comment:

The authors still misunderstand about prebiotic and probiotic concepts. Probiotics are referred to good microorganisms mostly lactic acid bacteria that when administered in an adequate amount, confer health benefits. Prebiotics, in general, are nutrient for probiotics. Prebiotic must resist to gastrointestinal system, thus passing through the GI tract to reach large intestinal where they can be fermentable by microbiota. The authors are always confused when expressing about prebiotic and probiotic concept. For example, Lines 32-34 the authors state that it (referring to grains) contains probiotic for enterobacteria species, …. which is not true.

The authors’ Answer:

Thanks the reviewer’s comment, I’ve carefully revised the error.

I’ve carefully rewrote the sentence. Please see page 1 line, 32-35.

  1. The reviewer’s comment:

The authors lacks basic in microbiology as I have previously mentioned but I have not seen any improvement in this manuscript. For example

Line 43, The most familiar lactobacilli are Lactobacillus, ……, and Enterococcus. However, Enterococcus is a cocci genus.

The authors’ Answer:

Thanks the reviewer’s comment, I’ve carefully revised the error.

The error has been revised. Please see page 2, line 44.

  1. The reviewer’s comment:

No information about how was L. brevis cultured? What kind of media and their. composition? What was the cultured conditions? And How did the author initially set up 9 log CFU/mL? Please add this information in section 2.1 which needs to be edited to “Materials, microorganism, and cultivation”

The authors’ Answer:

Thanks the reviewer’s comment, section 2.1 has been revised.

2.1 Materials, microorganism, and cultivation

This study used a lactic acid bacterium, L. brevis LUC 247, derived from Dr. Ying-Chen Lu’s strain bank at Chiayi University. Thaw the L. brevis LUC 247 strain from the freezer and add 0.1 mL to 0.9 mL of fresh deMan Rogosa and Sharpe (MRS) agar culture medium for activation. Incubate at 37°C for 24 h. Then, add the activated bacterial solution to a fresh MRS culture medium in a volume nine times larger than the bacterial solution. Incubate at 37°C for 24 h for the second activation and expansion.

The purity of the purchased reagents and chromatography standards from Sigma-Aldrich Chemical Co. (St. Louis, Missouri) was not lower than 98%.

Please see page 2 section 2.1 (line 77-85).

  1. The reviewer’s comment:

Section 2.2.1, I do not understand how the authors prepare bacterial suspension for dough making (line 86-89). Please rewrite to make it more understandable. 

Line 86, “The bacteria solution” Is it the bacterial culture?

Line 86, I do not understand “accounting for 1% of the total dough weight” what does it mean?

Line 88, “The supernatant was removed, and 4 mL of the bacteria solution was uniformly mixed with 200 mL of sterilized distilled water. Does it make sense? The authors removed supernatant and why was it retained 4 mL. 

The supernatant was removed, then the cell pellet was resuspended with 4 mL sterile water. The resulting cell suspension was mixed with 200 mL sterile water. Correct?

Line 96, remove "For various test measurement," and add more information. What were these samples analyzed? Total bacterial cell count, pH, total acid, organic acids, and so on.

Line 98, Is it for bread making? if so, please indicate in this section.

There is no methodology for section 3.6.

The authors’ Answer:

Thanks the reviewer’s comment. Section 2.2.1 has been revised.

To prepare the black rice sourdough, a bacterial suspension was combined with varying proportions of bread flour and black rice powder (seTable 1). The resulting mixtures were then fermented at 37℃ for 48 h, with samples taken at different intervals (0, 4, 8, 12, 24, 36, and 48 h) to analyze total bacterial cell count, pH, total acid, organic acids, antioxidant activities, and more. Once the optimal fermentation conditions were achieved, the black rice sourdough was freeze-dried for 48h. Then, it was pulverized to create type III sourdough flour and studied for its properties during storage.

please see page 2, section 2.2.1 (line 91-97) and Table 1.

  1. The reviewer’s comment:

Section 2.2.2, there are huge mistakes and misleading in this section.

Line 102, “The modification was performed …. (2019)” Does it make sense? Will. the readers understand the meaning that the authors would like to indicate? Please rewrite this paragraph. We do not say “aseptic saline solution”, “smearing for spread plate technique” in terms of microbiology. The condition stated in lines 105-106 does not develop the anaerobic condition at all !!!

The authors’ Answer:

Thanks the reviewer’s comment. Section 2.2.2 has been rewrite.

The lactic acid bacterial cell count was performed after the modification by referring to the method of Siepmann et al. (2019)[18]. In brief, The sourdough (0.1 g) was mixed with 0.9 mL of 0.85% sterile saline solution. The cell suspension was serially diluted with the identical diluent to make a 10-fold dilution. Then, 0.1 mL of the diluted cell suspension was spread on MRS agar. The plate was incubated at 37°C for 24 to 48 h. The viable cells were expressed as Log colony-forming units (CFU) per gram.

Please see page 3, Section 2.2.2 (line 106-111).

  1. The reviewer’s comment:

Section 2.2.3, I do not think that the authors modified method for pH measurement. This is too simple.

Line 113-116, how did the authors observed the end point of titration?

The authors’ Answer:

Thanks the reviewer’s comment, Section 2.2.3 has been revised.

The pH value determination was modified according to Coda et al. (2010) [31]. A desktop pH meter (MP220, Metter Toledo, UK, Switzerland) was calibrated with calibration solutions with pH values of 4.01 and 7.00. Next, 5 g of sourdough was mixed with 45 mL of distilled water, and measured the pH value.

The determination of titratable acid was modified based on Rühmkorf et al. (2012) [32]. Five grams of sourdough were mixed with 45 mL of distilled water and placed in a triangular flask. Next, 0.2 mL of 1% phenolphthalein indicator was added. The mixture was titrated with 0.1 N NaOH solution until the solution initially turned pink and did not fade within 0.5 min as the endpoint (pH=8.1~8.3). The consumed volume of the NaOH solution was then recorded upon reaching the endpoint of the titration, and the titration acid content was calculated.

please see page 3, section 2.2.3 (line 114-124).

  1. The reviewer’s comment:

Section 2.2.4, again, the sentence started with “A modification” which is not suitable.

Lines 125-127 contain incomplete sentence.

The authors’ Answer:

Thanks the reviewer’s comment, Section 2.2.3 has been revised.

The determination was performed after the modification by referring to the method of Clément et al. (2018) [22]. In brief, 1 g of sourdough was diluted in 19 mL of deionized (DI) water to achieve a twentyfold dilution. The mixture was then shaken for one minute and centrifuged at 12,000 rpm for 20 min. The supernatant was filtered using a 0.22 μm PVDF filter membrane and analyzed using high-performance liquid chromatography (HPLC) (SCL-10ADVP, Shimadzu, Kyoto, Japan). The analytical column was an Athena C18-WP (4.6 x 260 mm) (ANPEL, Shanghai, China). The mobile phase consisted of a mixture of acetonitrile and 0.05 M phosphoric acid solution (5:95 v/v), with a flow velocity of 0.6 mL/min and a column temperature of 30℃. The chromatogram was recorded at 318 nm. The fermentation quotient was calculated as the molar concentration ratio of lactic acid to acetic acid.

please see page 3, section 2.2.4 (line 127-137).

  1. The reviewer’s comment:

Section 2.2.10, again, the authors started the sentence with “A modification”

Which is not referred to methodology.

The authors’ Answer:

Thanks the reviewer’s comment, Section 2.2.10 has been revised.

The bread formula was modified using different proportions of bread flour and black rice sourdough powder, as Tafti et al. (2013) outlined. Four different proportions were used, including BB0 (no black rice sourdough powder), BB10 (10% black rice sourdough powder), BB20 (20% black rice sourdough powder), and BB30 (30% black rice sourdough powder). The ingredients used for the bread included bread flour, black rice sourdough powder, distilled water, yeast powder, salt, granulated sugar, milk powder, and salt-free cream. An intelligent automatic bread machine was used for production, which included mixing the ingredients at low speed for 3 min, stirring at medium speed for 10 min, keeping the mixture still for 15 min, stirring again at medium speed for 10 min, fermenting at 80% humidity for 27 min, reshaping slowly for 5 min, referencing at 80% humidity for 25 min, and finally baking at 150℃ for 1 h.

please see page 5, section 2.2.10 (line 211-221).

  1. The reviewer’s comment:

Section 3.1, Please avoid initiating the sentence with “Fig. shows” and check throughout the manuscript.

Please replace “bacteria count” by “bacterial cell count” and edit throughout the manuscript.

Please avoid describing growth of bacteria using “peak” it is not really scientific description.

Line 262-263, the authors misunderstand about probiotic “These nutrients can act as probiotics to …” which is not true !! Black rice bran contains various nutrients which some can act as prebiotics such as xylan and its oligosaccharides (xylooligosaccharides)

The authors’ Answer:

Thanks the reviewer’s comment, Section 3.1 has been revised.

Based on the growth curve of the lactic acid bacteria count shown in Figure 1(A), the bacterial counts of all groups increased rapidly during the first 12 h of fermentation. BS30 had the highest bacterial count among all groups, reaching 9.22 log CFU/g. The stationary phase of the bacterial count was observed between 12 and 36 h of fermentation, during which the nutrients in the black rice sourdough were almost depleted, leading to a dynamic equilibrium between living and dead bacterial counts. The bacterial count of BS60 started to decline after 24 h of fermentation. The bacterial counts of BS 0, BS15, BS30, BS45, and BS60 were 8.92, 8.18, 8.64, 7.99, and 7.84 CFU/g, respectively, at the end of 48 h of fermentation, indicating that the nutrients in the black rice sourdough were no longer sufficient to support the growth of the strains, causing cell death. Therefore, the optimal fermentation time for the black rice sourdough was between 12 and 36 h, during which the maximum bacterial count was observed. The dough's nutrient content and organic acid accumulation were critical factors in controlling the bacterial count. The lactic acid bacterial cell count was maintained at the peak point by making bread using the black rice sourdough in this fermentation interval. The strain viability was better, and more nutrient metabolites could be generated.

Starch degradation is an essential part of the sourdough fermentation process. The co-activation depends on the activity of amylase in the flour and the glucosidase in the species. After degradation, multiple fermentable sugars are generated to provide nutrients required by sourdough's lactic acid bacteria and yeast [18,38]. Black rice bran contains carbohydrates, proteins, oil, fat, and micronutrients. These nutrients can enhance lactic acid bacteria's growth and survival abilities [39].

Please see page 6 section 3.1 (line 247-268).

  1. The reviewer’s comment:

Section 3.2,

Line 281, Has L. brevis the strain that used in this study been claimed as probiotic? If not, please edit line 281 to “Lactic acid bacteria consume fermentable sugars”

Line 290, black rice does not contain many probiotics. It may contain some lactic acid bacteria. If this is not misleading, please provide related reference.

The authors’ Answer:

Thanks the reviewer’s comment, Section 3.2 has been revised.

The pH variation of the black rice sourdough with different proportions of black rice powder is shown in Figure 1(B). The pH values of all groups decreased rapidly from 0 to 24 h of fermentation, and the pH value decreased slowly after 24 h of fermentation. The pH value ranged from 3.77 to 3.83 after 48 h of fermentation.

The variation of the titratable acid content in the black rice sourdough is shown in Figure 1(C). The titratable acid of all groups increased at the highest rate after 8 to 36 h of fermentation, possibly because the lactic acid bacteria were in a logarithmic growth phase. During that time, the strains multiplied, the bacterial count remained high, and a large amount of organic acid was generated. Hence, the group with a higher proportion of black rice powder had a higher titratable acid content. The titratable acid increased slowly from 36 to 48 h because the lactic acid bacteria began to decline and die as the nutrition was exhausted. The pH value was negatively correlated with the titratable acid curve. Probiotics consume fermentable sugars as carbon sources for growth and generate organic acids. The organic acid enhances the protease and amylase activity in the flour, induces protein and starch degradation and provides more nutrient substances for the growth of lactic acid bacteria [38]. The higher the additional level of black rice powder, the greater the yield of organic acid, as black rice contains a great fermentable carbohydrates. The pH value is a critical factor in the survival of lactic acid bacteria prebiotics, and acid accumulation during fermentation may be the reason for the decrease in lactic acid bacterial counts. Pereira et al. (2017) [40] indicated that a pH above 4.0 would not influence lactic acid bacteria, but a pH below 4.0 might inhibit their existence. Therefore, decrease in lactic acid bacterial cell count might have been caused by inhibition regarding due to excessive accumulation of organic acid. The lactic acid bacteria count of the BS60 decreased after 36 h and might have been inhibited earlier than the other groups due to its higher titratable acidity.

Please see page 6 section 3.2 (line 271-295).

Please edit Figure 1, Figure 2, and Figure 3 by extending the font size of axis titles and legend as they are too small.

The authors’ Answer:

Thanks the reviewer’s comment, All figures have been extended the font size.

please Figure 1, Figure 2, and Figure 3

Please look at the attachment for other comments

The authors’ Answer:

Thanks the reviewer’s comment, the other comments have been revised.

Many typographical errors and abbreviations have been revised. All the lines and pages indicated above are in the revised manuscript. We hope that all these changes fulfill the requirements to make the manuscript acceptable for publication in Journal of Food and Drug analysis.

Looking forward to hearing from you soon.

Sincerely yours,

Chih-Wei Chen

Reviewer 2 Report (New Reviewer)

The English language and writing style of the manuscript need to be improved. The storage study is incomplete. For a product development, it is recommended to include microbiological, physicochemical and sensory test to determine the shelf life quality. 

Author Response

03/03/23

Prof. Dr. Arun K. Bhunia
Editor-in-Chief
Foods

Dear Professor,

Thank you for considering the Resubmit of our manuscript: Foods-2258232, entitled " Development and optimization of black rice (Oryza sativa L.) sourdough bread fermented by Levilactobacillus brevis LUC 247 for physicochemical characteristics, and antioxidant capacity", by Lai. et al. for publication in Foods. We are thankful to the referees and the Editor for pointing out some important modifications needed in the report. We believe the comments have been highly constructive and useful in restructuring the manuscript. We have thoughtfully taken into account these comments. The explanation of what we have changed in response to the reviewers’ concerns is given point by point in the following pages.

Review 1

Comments and Suggestions for Authors

Dear Authors, 

After the evaluation, the authors lack expertise in the microbiology field. Once again, I found so many mistakes in the manuscript, particularly English writing in part of microbiology which is not only limited in methodology but also results and discussion. Previously, I commented that this manuscript needs English editing and proofreading prior to submission. In this submission, however, the authors did not properly respond to my request. The manuscript was prepared with careless consideration and concern, especially to my comments. There are grammatical errors, incomplete sentences, and misleading sentences throughout the manuscript. 

  1. The reviewer’s comment:

The authors still misunderstand about prebiotic and probiotic concepts. Probiotics are referred to good microorganisms mostly lactic acid bacteria that when administered in an adequate amount, confer health benefits. Prebiotics, in general, are nutrient for probiotics. Prebiotic must resist to gastrointestinal system, thus passing through the GI tract to reach large intestinal where they can be fermentable by microbiota. The authors are always confused when expressing about prebiotic and probiotic concept. For example, Lines 32-34 the authors state that it (referring to grains) contains probiotic for enterobacteria species, …. which is not true.

The authors’ Answer:

Thanks the reviewer’s comment, I’ve carefully revised the error.

I’ve carefully rewrote the sentence. Please see page 1 line, 32-35.

  1. The reviewer’s comment:

The authors lacks basic in microbiology as I have previously mentioned but I have not seen any improvement in this manuscript. For example

Line 43, The most familiar lactobacilli are Lactobacillus, ……, and Enterococcus. However, Enterococcus is a cocci genus.

The authors’ Answer:

Thanks the reviewer’s comment, I’ve carefully revised the error.

The error has been revised. Please see page 2, line 44.

  1. The reviewer’s comment:

No information about how was L. brevis cultured? What kind of media and their. composition? What was the cultured conditions? And How did the author initially set up 9 log CFU/mL? Please add this information in section 2.1 which needs to be edited to “Materials, microorganism, and cultivation”

The authors’ Answer:

Thanks the reviewer’s comment, section 2.1 has been revised.

2.1 Materials, microorganism, and cultivation

This study used a lactic acid bacterium, L. brevis LUC 247, derived from Dr. Ying-Chen Lu’s strain bank at Chiayi University. Thaw the L. brevis LUC 247 strain from the freezer and add 0.1 mL to 0.9 mL of fresh deMan Rogosa and Sharpe (MRS) agar culture medium for activation. Incubate at 37°C for 24 h. Then, add the activated bacterial solution to a fresh MRS culture medium in a volume nine times larger than the bacterial solution. Incubate at 37°C for 24 h for the second activation and expansion.

The purity of the purchased reagents and chromatography standards from Sigma-Aldrich Chemical Co. (St. Louis, Missouri) was not lower than 98%.

Please see page 2 section 2.1 (line 77-85).

  1. The reviewer’s comment:

Section 2.2.1, I do not understand how the authors prepare bacterial suspension for dough making (line 86-89). Please rewrite to make it more understandable. 

Line 86, “The bacteria solution” Is it the bacterial culture?

Line 86, I do not understand “accounting for 1% of the total dough weight” what does it mean?

Line 88, “The supernatant was removed, and 4 mL of the bacteria solution was uniformly mixed with 200 mL of sterilized distilled water. Does it make sense? The authors removed supernatant and why was it retained 4 mL. 

The supernatant was removed, then the cell pellet was resuspended with 4 mL sterile water. The resulting cell suspension was mixed with 200 mL sterile water. Correct?

Line 96, remove "For various test measurement," and add more information. What were these samples analyzed? Total bacterial cell count, pH, total acid, organic acids, and so on.

Line 98, Is it for bread making? if so, please indicate in this section.

There is no methodology for section 3.6.

The authors’ Answer:

Thanks the reviewer’s comment. Section 2.2.1 has been revised.

To prepare the black rice sourdough, a bacterial suspension was combined with varying proportions of bread flour and black rice powder (seTable 1). The resulting mixtures were then fermented at 37℃ for 48 h, with samples taken at different intervals (0, 4, 8, 12, 24, 36, and 48 h) to analyze total bacterial cell count, pH, total acid, organic acids, antioxidant activities, and more. Once the optimal fermentation conditions were achieved, the black rice sourdough was freeze-dried for 48h. Then, it was pulverized to create type III sourdough flour and studied for its properties during storage.

please see page 2, section 2.2.1 (line 91-97) and Table 1.

  1. The reviewer’s comment:

Section 2.2.2, there are huge mistakes and misleading in this section.

Line 102, “The modification was performed …. (2019)” Does it make sense? Will. the readers understand the meaning that the authors would like to indicate? Please rewrite this paragraph. We do not say “aseptic saline solution”, “smearing for spread plate technique” in terms of microbiology. The condition stated in lines 105-106 does not develop the anaerobic condition at all !!!

The authors’ Answer:

Thanks the reviewer’s comment. Section 2.2.2 has been rewrite.

The lactic acid bacterial cell count was performed after the modification by referring to the method of Siepmann et al. (2019)[18]. In brief, The sourdough (0.1 g) was mixed with 0.9 mL of 0.85% sterile saline solution. The cell suspension was serially diluted with the identical diluent to make a 10-fold dilution. Then, 0.1 mL of the diluted cell suspension was spread on MRS agar. The plate was incubated at 37°C for 24 to 48 h. The viable cells were expressed as Log colony-forming units (CFU) per gram.

Please see page 3, Section 2.2.2 (line 106-111).

  1. The reviewer’s comment:

Section 2.2.3, I do not think that the authors modified method for pH measurement. This is too simple.

Line 113-116, how did the authors observed the end point of titration?

The authors’ Answer:

Thanks the reviewer’s comment, Section 2.2.3 has been revised.

The pH value determination was modified according to Coda et al. (2010) [31]. A desktop pH meter (MP220, Metter Toledo, UK, Switzerland) was calibrated with calibration solutions with pH values of 4.01 and 7.00. Next, 5 g of sourdough was mixed with 45 mL of distilled water, and measured the pH value.

The determination of titratable acid was modified based on Rühmkorf et al. (2012) [32]. Five grams of sourdough were mixed with 45 mL of distilled water and placed in a triangular flask. Next, 0.2 mL of 1% phenolphthalein indicator was added. The mixture was titrated with 0.1 N NaOH solution until the solution initially turned pink and did not fade within 0.5 min as the endpoint (pH=8.1~8.3). The consumed volume of the NaOH solution was then recorded upon reaching the endpoint of the titration, and the titration acid content was calculated.

please see page 3, section 2.2.3 (line 114-124).

  1. The reviewer’s comment:

Section 2.2.4, again, the sentence started with “A modification” which is not suitable.

Lines 125-127 contain incomplete sentence.

The authors’ Answer:

Thanks the reviewer’s comment, Section 2.2.3 has been revised.

The determination was performed after the modification by referring to the method of Clément et al. (2018) [22]. In brief, 1 g of sourdough was diluted in 19 mL of deionized (DI) water to achieve a twentyfold dilution. The mixture was then shaken for one minute and centrifuged at 12,000 rpm for 20 min. The supernatant was filtered using a 0.22 μm PVDF filter membrane and analyzed using high-performance liquid chromatography (HPLC) (SCL-10ADVP, Shimadzu, Kyoto, Japan). The analytical column was an Athena C18-WP (4.6 x 260 mm) (ANPEL, Shanghai, China). The mobile phase consisted of a mixture of acetonitrile and 0.05 M phosphoric acid solution (5:95 v/v), with a flow velocity of 0.6 mL/min and a column temperature of 30℃. The chromatogram was recorded at 318 nm. The fermentation quotient was calculated as the molar concentration ratio of lactic acid to acetic acid.

please see page 3, section 2.2.4 (line 127-137).

  1. The reviewer’s comment:

Section 2.2.10, again, the authors started the sentence with “A modification”

Which is not referred to methodology.

The authors’ Answer:

Thanks the reviewer’s comment, Section 2.2.10 has been revised.

The bread formula was modified using different proportions of bread flour and black rice sourdough powder, as Tafti et al. (2013) outlined. Four different proportions were used, including BB0 (no black rice sourdough powder), BB10 (10% black rice sourdough powder), BB20 (20% black rice sourdough powder), and BB30 (30% black rice sourdough powder). The ingredients used for the bread included bread flour, black rice sourdough powder, distilled water, yeast powder, salt, granulated sugar, milk powder, and salt-free cream. An intelligent automatic bread machine was used for production, which included mixing the ingredients at low speed for 3 min, stirring at medium speed for 10 min, keeping the mixture still for 15 min, stirring again at medium speed for 10 min, fermenting at 80% humidity for 27 min, reshaping slowly for 5 min, referencing at 80% humidity for 25 min, and finally baking at 150℃ for 1 h.

please see page 5, section 2.2.10 (line 211-221).

  1. The reviewer’s comment:

Section 3.1, Please avoid initiating the sentence with “Fig. shows” and check throughout the manuscript.

Please replace “bacteria count” by “bacterial cell count” and edit throughout the manuscript.

Please avoid describing growth of bacteria using “peak” it is not really scientific description.

Line 262-263, the authors misunderstand about probiotic “These nutrients can act as probiotics to …” which is not true !! Black rice bran contains various nutrients which some can act as prebiotics such as xylan and its oligosaccharides (xylooligosaccharides)

The authors’ Answer:

Thanks the reviewer’s comment, Section 3.1 has been revised.

Based on the growth curve of the lactic acid bacteria count shown in Figure 1(A), the bacterial counts of all groups increased rapidly during the first 12 h of fermentation. BS30 had the highest bacterial count among all groups, reaching 9.22 log CFU/g. The stationary phase of the bacterial count was observed between 12 and 36 h of fermentation, during which the nutrients in the black rice sourdough were almost depleted, leading to a dynamic equilibrium between living and dead bacterial counts. The bacterial count of BS60 started to decline after 24 h of fermentation. The bacterial counts of BS 0, BS15, BS30, BS45, and BS60 were 8.92, 8.18, 8.64, 7.99, and 7.84 CFU/g, respectively, at the end of 48 h of fermentation, indicating that the nutrients in the black rice sourdough were no longer sufficient to support the growth of the strains, causing cell death. Therefore, the optimal fermentation time for the black rice sourdough was between 12 and 36 h, during which the maximum bacterial count was observed. The dough's nutrient content and organic acid accumulation were critical factors in controlling the bacterial count. The lactic acid bacterial cell count was maintained at the peak point by making bread using the black rice sourdough in this fermentation interval. The strain viability was better, and more nutrient metabolites could be generated.

Starch degradation is an essential part of the sourdough fermentation process. The co-activation depends on the activity of amylase in the flour and the glucosidase in the species. After degradation, multiple fermentable sugars are generated to provide nutrients required by sourdough's lactic acid bacteria and yeast [18,38]. Black rice bran contains carbohydrates, proteins, oil, fat, and micronutrients. These nutrients can enhance lactic acid bacteria's growth and survival abilities [39].

Please see page 6 section 3.1 (line 247-268).

  1. The reviewer’s comment:

Section 3.2,

Line 281, Has L. brevis the strain that used in this study been claimed as probiotic? If not, please edit line 281 to “Lactic acid bacteria consume fermentable sugars”

Line 290, black rice does not contain many probiotics. It may contain some lactic acid bacteria. If this is not misleading, please provide related reference.

The authors’ Answer:

Thanks the reviewer’s comment, Section 3.2 has been revised.

The pH variation of the black rice sourdough with different proportions of black rice powder is shown in Figure 1(B). The pH values of all groups decreased rapidly from 0 to 24 h of fermentation, and the pH value decreased slowly after 24 h of fermentation. The pH value ranged from 3.77 to 3.83 after 48 h of fermentation.

The variation of the titratable acid content in the black rice sourdough is shown in Figure 1(C). The titratable acid of all groups increased at the highest rate after 8 to 36 h of fermentation, possibly because the lactic acid bacteria were in a logarithmic growth phase. During that time, the strains multiplied, the bacterial count remained high, and a large amount of organic acid was generated. Hence, the group with a higher proportion of black rice powder had a higher titratable acid content. The titratable acid increased slowly from 36 to 48 h because the lactic acid bacteria began to decline and die as the nutrition was exhausted. The pH value was negatively correlated with the titratable acid curve. Probiotics consume fermentable sugars as carbon sources for growth and generate organic acids. The organic acid enhances the protease and amylase activity in the flour, induces protein and starch degradation and provides more nutrient substances for the growth of lactic acid bacteria [38]. The higher the additional level of black rice powder, the greater the yield of organic acid, as black rice contains a great fermentable carbohydrates. The pH value is a critical factor in the survival of lactic acid bacteria prebiotics, and acid accumulation during fermentation may be the reason for the decrease in lactic acid bacterial counts. Pereira et al. (2017) [40] indicated that a pH above 4.0 would not influence lactic acid bacteria, but a pH below 4.0 might inhibit their existence. Therefore, decrease in lactic acid bacterial cell count might have been caused by inhibition regarding due to excessive accumulation of organic acid. The lactic acid bacteria count of the BS60 decreased after 36 h and might have been inhibited earlier than the other groups due to its higher titratable acidity.

Please see page 6 section 3.2 (line 271-295).

Please edit Figure 1, Figure 2, and Figure 3 by extending the font size of axis titles and legend as they are too small.

The authors’ Answer:

Thanks the reviewer’s comment, All figures have been extended the font size.

please Figure 1, Figure 2, and Figure 3

Please look at the attachment for other comments

The authors’ Answer:

Thanks the reviewer’s comment, the other comments have been revised.

Many typographical errors and abbreviations have been revised. All the lines and pages indicated above are in the revised manuscript. We hope that all these changes fulfill the requirements to make the manuscript acceptable for publication in Journal of Food and Drug analysis.

Looking forward to hearing from you soon.

Sincerely yours,

Chih-Wei Chen

Round 2

Reviewer 1 Report (Previous Reviewer 1)

Dear authors,

Thank you for your response. Most comments have now been completed. However, please complete some minor questions according to the attachment. Please also reconsider modifying the title of the article with regard to my comment.

Author Response

08/03/23

Prof. Dr. Arun K. Bhunia
Editor-in-Chief
Foods

Dear Professor,

Thank you for considering the Resubmit of our manuscript: Foods-2258232, entitled " Development and optimization of black rice (Oryza sativa L.) sourdough fermented by Levilactobacillus brevis LUC 247 for physicochemical characteristics, and antioxidant capacity", by Lai. et al. for publication in Foods. We thank the referees and the Editor for highlighting some important modifications needed in the report. We believe the comments have been highly constructive and useful in restructuring the manuscript. We have thoughtfully considered these comments. The explanation of what we have changed in response to the reviewers’ concerns is given point in the following pages.

Review 1

Comments and Suggestions for Authors

Thank you for your response. Most comments have now been completed. However, please complete some minor questions according to the attachment. Please also reconsider modifying the title of the article with regard to my comment.

The authors’ Answer:

Thanks the reviewer’s comment, I’ve carefully revised the error and also modified the title of the article with regard to the review comment.

Many typographical errors and abbreviations have been revised. All the lines and pages indicated above are in the revised manuscript. We hope all these changes fulfill the requirements to make the manuscript acceptable for publication in the Foods

Looking forward to hearing from you soon.

Sincerely yours,

Chih-Wei Chen

This manuscript is a resubmission of an earlier submission. The following is a list of the peer review reports and author responses from that submission.

Round 1

Reviewer 1 Report

The manuscript foods-2097200 entitled “Fermentation by Levilactobacillus brevis LUC 247 on physicochemical characteristics and preservation on type III sourdough and black rice (Oryza sativa L.) sourdough bread” described production of type III sourdough  and black rice sourdough for bread making and evaluation of the sourdoughs characteristics including microbiological, biochemical, and chemical properties. Furthermore, bread was made using the different sourdoughs and was further determined its physical properties and storage time. Overview, the concept, experimental design, and presentation of this manuscript are interesting to readers, however, the English writing is very poor and difficult to understand and is further required substantial improvement before publication. It is now not qualified enough to publish in Foods. I recommended to rewrite the manuscript and resubmitted to this issue if it is possible. This manuscript needs writers who have expertise in microbiology and biotechnology besides food science. Following are the overview major comments.

1. This manuscript lacks background stated in the introduction. The authors need to describe why this research work is established; merits of this work compared to the previous publications.

2. Please check the definition and concept of prebiotics and probiotics. Please look at lines 30, 33, and line 238, how did the authors think about probiotics?

3. Please check how to write scientific name of microorganisms.

4. The title of this manuscript needs revision.

5. More information is required in some methods such as

              - what kind of HPLC column did the authors use for analysis of organic acids?

              - how did the authors determine antioxidant activity? (not just referring to the previous works)

              - in 2.2.6, how did the authors extract sourdough for further determination of total polyphenol and total anthocyanin contents

6. The citation in text of this manuscript should be edited into the correct form.

7. In the results section, the authors always compared values without direction (higher or lower) such as lines 285, 305, 313

8. The conclusion should be concise.

Reviewer 2 Report

This manuscript by Syue-Fong Lia et al. investigates utilization of Levilactobacillus brevis strain LUC 247 for production of type III sourdough from black rice powder with subsequent backing of black rice sourdough bread. The topic of the article meets the scope of the “Foods” and I believe that it will be of interest to the readers. Yet, I’ve met some substantial flaws and shortcomings in the manuscript and suppose that the paper needs major revision.

The manuscript (starting from the very title) is not optimal regarding the quality of the English and needs revision to make the text more intelligible.

Major Critiques:

1. Is the strain L. brevis LUC 247 available from Dr. Ying-Chen Lu’s strain bank at Chiayi University? If not, the results of the paper have no practical application.

2. Lines 75 and 249: which strainS? Here and throughout the manuscript authors write about multiple strains and LAB (e.g. title of figure 1). There is an inconsistency, because according to Materials and methods section only one strain was used for fermentation. If authors suggest that still the fermentation investigated in the paper was fulfilled by multiple bacterial strains (so do I, because bread flour and black rice powder were not sterile), then the method used for enumeration of bacteria (Fig. 1A) is not applicable. Please, clarify this question about microbial content of sourdough or make necessary corrections in the text to demonstrate the unique presence of L. brevis strain LUC 247.

 3. Yeast powder was added to the mixture for bread (line 185). I think that in this case the product cannot be called a “black rice sourdough bread”. In this case physical and other properties of bread are not due to the black rice sourdough, but may originate from alcoholic fermentation fulfilled by yeast.

4. Sentence “Additionally, it contains…” (lines 32-35): several substantive and grammar mistakes. Here and throughout the manuscript wrong usage of term “probiotics” (instead of “prebiotics”). Line 39: Dietary fiber consists of nondigestible carbohydrates (discussed in line 32). It would be logic to combine this information.

5. Line 62: not “of the lactic acid bacteria.”, but sourdough. Lines 63-64: Temperature is usually not discussed, but determined.

 6. Lines 143-144 and reference 25: of Reagent set al., (2005) [25] - ? Correct reference 25 in the text and in reference list.

7. Line 167: how is it connected to TEA?

Minor Critiques:

Line 42: 380 kinds of bacteria → species of bacteria?

The genera in line 43 and L. brevis throughout the manuscript should be italicized.

Line 46: several kinds of organic acid → of organic acids

Lines 49-50: wrong phrase (can catabolize carbohydrates by Kyoto encyclopedia of genes and genomes). By KEGG → According to KEGG, but actually these properties of L. brevis are described in many sourses, e.g. Bergey Manual.

Line 51: microflora → microbiota

Line 173: The sample was measured water activity and... - ? (English correction is required).

Figure 2: OX axis labeling is not readable.

Figure 3: Letters a,b,c meaning is not identified.